# Influence of Brain-Derived Neurotrophic Factor Genotype on Short-Latency Afferent Inhibition and Motor Cortex Metabolites

**DOI:** 10.3390/brainsci11030395

**Published:** 2021-03-20

**Authors:** Ryoki Sasaki, Naofumi Otsuru, Shota Miyaguchi, Sho Kojima, Hiraku Watanabe, Ken Ohno, Noriko Sakurai, Naoki Kodama, Daisuke Sato, Hideaki Onishi

**Affiliations:** 1Institute for Human Movement and Medical Sciences, Niigata University of Health and Welfare, Niigata City, Niigata 950-3198, Japan; otsuru@nuhw.ac.jp (N.O.); miyaguchi@nuhw.ac.jp (S.M.); kojima@nuhw.ac.jp (S.K.); hpm19011@nuhw.ac.jp (H.W.); kodama@nuhw.ac.jp (N.K.); daisuke@nuhw.ac.jp (D.S.); onishi@nuhw.ac.jp (H.O.); 2Discipline of Physiology, Adelaide Medical School, The University of Adelaide, Adelaide 5000, Australia; 3Department of Physical Therapy, Niigata University of Health and Welfare, Niigata City, Niigata 950-3198, Japan; 4Department of Radiological Technology, Niigata University of Health and Welfare, Niigata City, Niigata 950-3198, Japan; ken-ohno@nuhw.ac.jp (K.O.); noriko-sakurai@nuhw.ac.jp (N.S.); 5Department of Health and Sports, Niigata University of Health and Welfare, Niigata City, Niigata 950-3198, Japan

**Keywords:** brain-derived neurotrophic factor genotype, excitatory neurometabolite, inhibitory network, primary motor cortex, short-latency afferent inhibition

## Abstract

The Met allele of the brain-derived neurotrophic factor (BDNF) gene confers reduced cortical BDNF expression and associated neurobehavioral changes. BDNF signaling influences the survival, development, and synaptic function of cortical networks. Here, we compared gamma-aminobutyric acid (GABA)ergic network activity in the human primary motor cortex (M1) between the Met (Val/Met and Met/Met) and non-Met (Val/Val) genotype groups. Short- and long-interval intracortical inhibition, short-latency afferent inhibition (SAI), and long-latency afferent inhibition were measured using transcranial magnetic stimulation (TMS) as indices of GABAergic activity. Furthermore, the considerable inter-individual variability in inhibitory network activity typically measured by TMS may be affected not only by GABA but also by other pathways, including glutamatergic and cholinergic activities; therefore, we used 3-T magnetic resonance spectroscopy (MRS) to measure the dynamics of glutamate plus glutamine (Glx) and choline concentrations in the left M1, left somatosensory cortex, and right cerebellum. All inhibitory TMS conditions produced significantly smaller motor-evoked potentials than single-pulses. SAI was significantly stronger in the Met group than in the Val/Val group. Only the M1 Glx concentration was significantly lower in the Met group, while the BDNF genotype did not affect choline concentration in any region. Further, a positive correlation was observed between SAI and Glx concentrations only in M1. Our findings provide evidence that the BDNF genotype regulates both the inhibitory and excitatory circuits in human M1. In addition, lower Glx concentration in the M1 of Met carriers may alter specific inhibitory network on M1, thereby influencing the cortical signal processing required for neurobehavioral functions.

## 1. Introduction

The gamma-aminobutyric acid (GABA)-mediated inhibitory network of the primary motor cortex (M1) is critical for motor coordination. Paired-pulse paradigms delivered non-invasively via transcranial magnetic stimulation (TMS) can evaluate various forms of the local GABAergic activity in the human M1. These stimulus paradigms include short-interval intracortical inhibition (SICI) evoked by inter-stimulus intervals (ISIs) of 1–5 ms, long-interval intracortical inhibition (LICI) evoked by ISIs of 100–150 ms [1,2], and combined TMS with peripheral electrical stimulation at ISIs of 20–30 ms to elicit short-latency afferent inhibition (SAI) or ISIs of 200–1000 ms to elicit long-latency afferent inhibition (LAI) [3,4]. This M1-driven inhibitory network activity is modulated by multiple additional factors, resulting in substantial inter-individual variability [5]. The brain-derived neurotrophic factor (BDNF) genotype may be one of the factors that cause the variability of the gamma-aminobutyric acid (GABA)ergic inhibitory network. BDNF, a member of the neurotrophic family, is released throughout the brain and plays a fundamental role in the survival, maintenance, and growth of neurons [6]. The atypical BDNF genotype, such as Val/Met and Met/Met, are often observed to be associated with a reduction in BDNF secretion [7], a decrease in M1 plasticity [8], and poor motor learning [9]. An abnormal BDNF secretion may be involved in the pathogenesis or the clinical onset of neurological [10], psychiatric [11], and immune-related diseases in humans [12]. At the neural level, an atypical BDNF genotype also confers weaker GABA receptor-mediated synaptic transmission to mouse pyramidal neurons [13] and NMDA receptor-mediated glutamatergic activity in the mouse hippocampus [14]. Therefore, these modifications may contribute to inhibitory network abnormalities in M1 and associated neurobehavioral deficits. However, the influence of BDNF genotype on neurotransmission and neurometabolism has not been examined in the human M1.

If the inhibitory network defined using TMS measurements depends on the BDNF genotype, it may ultimately be associated with differences in neurotransmitter such as GABA within M1 rather than altered excitatory neurotransmission. However, recent studies that used magnetic resonance spectroscopy (MRS), which is a reliable means of directly measuring the concentrations of neurometabolites, reported no relation between GABA concentration in sensorimotor cortex and either SICI or LICI [15,16,17,18] despite these inhibitory TMS conditions are strongly influenced by GABAergic activity [19]. There are two potential explanations for this discrepancy. First, MRS and TMS techniques may reflect different aspect of GABAergic activity. MRS directly measures the entire pool of a given neurometabolite, including metabolic, intracellular, and extracellular fractions [17]. On the other hand, paired-pulse TMS indirectly evaluates the strength of inhibition (compared to single-pulses), and reflects the dynamic synaptic activation of corticospinal cells [20]. This suggests that GABAergic activity as assessed by MRS may not be strongly associated with TMS-assessed GABAergic activity. In contrast, Dyke et al., (2017) [17] observed the correlations between MRS-assessed concentrations of the primary excitatory neurometabolite glutamate (Glu), and both the TMS-assessed input/output curve plateau and intracortical facilitation. Furthermore, Tremblay et al. (2013) [18] also reported a correlation between the cortical silent period duration including GABAergic activity as assessed by TMS and MRS-assessed Glx (the combination of Glu and glutamine) but not GABA. These observations can explain that excitatory neurotransmitters contribute to TMS-induced inhibitory network activity by modulating the excitability of GABA neurons or by counteracting the postsynaptic effects of GABAergic transmission [21,22], suggesting that TMS-assessed activity actually reflects both GABAergic and glutaminergic activities. Although it is unclear whether MRS-measured neurometabolite concentrations reflect synaptic activation by TMS, it is apparent that MRS-assessed excitatory neurometabolite reflects more TMS measurements than MRS-assessed GABA. Therefore, any BDNF genotype effects on inhibitory cortical network activity as measured by TMS may actually reflect altered glutamatergic activity and may be associated with excitatory neurometabolite concentrations rather than GABA concentrations, as assessed by MRS. 

The influence of the BDNF Met allele on the inhibitory and excitatory circuits in the human M1 has not been investigated with TMS and MRS. Using both TMS and MRS would provide with unique information from different perspectives. In addition, to elucidating BDNF-related changes in inhibitory and excitatory circuits on M1 may help understand how the BDNF genotype influences behavioral changes. In the present study, we investigated three hypotheses: (1) BDNF genotype affects TMS-induced inhibitory networks as evidenced by changes in SICI, LICI, SAI, and/or LAI; (2) BDNF genotype influences cortical excitatory neurometabolite concentrations; (3) there is a relationship between TMS-induced inhibitory networks and M1 excitatory neurometabolites. To test this hypothesis, we also analyzed neurometabolite concentrations in the primary somatosensory cortex (S1) and cerebellum in addition to M1 because motor-evoked potentials (MEPs) reduction induced by somatosensory inputs may affect the ways in which S1 and cerebellum receive these signals. Collectively, these findings may suggest that the BDNF Met genotype disrupts M1 function through the inhibitory and excitatory networks.

## 2. Materials and Methods

### 2.1. Participants

A total of 58 right-handed healthy participants [36 males and 22 females; age (mean ± SD): 22.0 ± 1.8 years, range: 20–31 years] without a history of neurological or psychiatric disorders were recruited for this study. Participants participated in the TMS experiment [33 males and 12 females; age: 22.1 ± 2.0 (20–31) years], MRS experiment [15 males and 15 females; age: 22.1 ± 1.7 (21–29) years]. Seventeen participants were overlapping in both experiments [12 males and 5 females; age: 21.7 ± 1.7 (21–29) years]. A total of 13 participants who participated in the magnetic resonance (MR) experiment did not participate in the TMS experiment. TMS was performed in accordance with the current TMS safety guidelines [23] by an experimenter with 5 years of TMS experience. This study conformed to the Declaration of Helsinki and was approved by the ethics committee of Niigata University of Health and Welfare. Each subject provided written informed consent prior to participation.

### 2.2. Genetic Analysis

All participants underwent blood sample collection and genomic DNA extraction, and the samples were analyzed based on the SNP database (BDNF-rs6265) of the National Center for Biotechnology Information. Polymerase chain reaction was used for BDNF genotyping as previously described [24]. Participants with the Val/Met or Met/Met genotype were merged into a single Met group for the MRS experiment owing to the small number of Met/Met carriers.

### 2.3. Electromyography (EMG)

The EMG was recorded using disposable Ag/AgCl electrodes placed over the right first dorsal interosseous muscle in a belly−tendon montage. Recordings were sampled at 4000 Hz using an A/D converter (Power Lab 8/30, AD Instruments, Colorado Springs, CO, USA), amplified 100× (A-DL-720-140, 4 Assist, Tokyo, Japan), and band-pass filtered between 20 and 1000 Hz.

### 2.4. TMS Settings

TMS pulses were delivered through a figure-of-eight coil (9.5 cm diameter) connected to two Magstim 200 stimulators (Magstim, Dyfed, UK). The coil was oriented at an angle of 45° to induce a posterior-to-anterior current over the left M1 at the location producing the largest and most consistent MEPs. The individual position and orientation of the coil were digitally registered according to the structural magnetic resonance images of a template using Visor2 TMS Neuronavigation System (Eemagine Medical Imaging Solutions GmbH, Berlin, Germany). TMS intensity was adjusted to evoke an MEP amplitude of approximately 1 mV peak-to-peak in the first dorsal interosseous muscle. The resting motor threshold (RMT) was determined as the minimum stimulus intensity that elicited MEPs > 50 µV in a minimum 5 of the 10 trials.

### 2.5. Inhibitory TMS Conditions

The M1 inhibitory network was evaluated using paired-pulse paradigms and single TMS pulses paired with peripheral electrical stimulation. The test stimulus intensity was set to evoke approximately 1.0 mV MEP amplitudes from single-pulse TMS. For SICI, the conditioning stimulus intensity was set at 80% RMT with an ISI of 2 ms [1]. For LICI, the conditioning stimulus intensity was set to evoke approximately 1 mV MEPs with an ISI of 100 ms [25]. To assess SAI and LAI, the right ulnar nerve was electrically stimulated at the wrist at 110% of the M-wave threshold. The two ISIs separating electrical stimulation and TMS were set to 20 (SAI_1) and 25 ms (SAI_2) because different ISIs may rely on cerebellum- or somatosensory cortex-mediated M1 pathways according to a paired associative stimulation study [26]. A 200 ms ISI was used for LAI [27].

### 2.6. MR Data Acquisition

Magnetic resonance data were acquired using a 3 T Vantage Galan MRI scanner (Canon Medical Systems, Tochigi, Japan) with a 32-channel head SPEEDER coil. Anatomical images were acquired using a T1-weighted 3D magnetization-prepared rapid gradient echo sequence with the following parameters: repetition time (TR) = 5.8 ms, echo time (TE) = 2.7 ms, flip angle = 9°, inversion time (TI) = 900 ms, slice thickness = 1.2 mm, field of view = 23 × 23 cm^2^, acquisition matrix = 256 × 256 mm^2^, number of slices = 160, and slice gap = non-gap.

Functional (f)MRI was performed to confirm the location of the M1 hand area using the following echo planar imaging sequence: TR = 2000 ms, TE = 25 ms, flip angle = 85°, slice thickness = 3 mm, field of view = 24 × 24 cm^2^, acquisition matrix = 64 × 64 mm^2^, number of slices = 34, and slice gap = 1 mm. Participants were instructed to repeatedly tap their right index finger at a frequency of 1.0 Hz. Resting and finger tapping conditions were alternated for 30 s each in three blocks (1 block = 60 s). Three recording spans at the beginning of data acquisition were removed from the analysis owing to the unstable signal. Maximum activation of the left M1 was determined by analyzing the BOLD signal using the M-power V5.0 MRS application prior to MRS recording. The S1 hand area was identified based on the location of the left M1 hand area and central sulcus. The position of the right cerebellum was set at the top of the right cerebellum corresponding to the upper voxel. 

MRS data were acquired using a point resolved spectroscopy sequence (PRESS) with the following parameter settings: TR = 2000 ms, TE = 32 ms, 128 averages for each region, voxels of interest over left M1 and left S1 covering 15 × 15 × 15 mm^3^, voxels of interest over the right cerebellum covering 10 × 20 × 20 mm^3^, and chemical shift selective saturation to suppress the water signal. The exclusion criteria for data were as follows: full width at half maximum (FWHM) > 15 Hz for M1 and S1 voxels and 30 Hz for cerebellum voxel.

### 2.7. Experimental Protocol

Figure 1 illustrates the experimental protocol. Blood samples were collected from all participants before the TMS and MRS experiments, and the two experiments were performed on separate days. Magnetic resonance images (i.e., structural MRI, fMRI, and MRS) were recorded on the same day in the supine position and in the following order: structural MRI, fMRI, MRS. MRS data were recorded after M1 voxel detection based on fMRI data.

For TMS experiments, participants sat in a comfortable reclining chair with a mounted headrest. A total of 40 single-pulse MEPs and inhibitory TMS conditions were measured in a randomized order with an inter-trial interval of 4–6 s. Forty MEPs per condition were collected over two TMS sessions (1 session = 20 MEPs per condition) separated by approximately 10 min. 

### 2.8. Analysis of TMS Data

LabChart8 software (AD Instruments) was used for MEP analysis. Trials were rejected when the root mean square background EMG activity exceeded 30 µV from −100 ms before the TMS trigger to the end of the MEP waveform. The peak-to-peak amplitudes of single-pulse MEPs and paired MEPs in each inhibitory TMS condition were averaged for each participant within session 1, within session 2, and across both sessions. The MEP amplitudes for the inhibitory TMS conditions were normalized to single-pulse MEPs.

### 2.9. Analysis of MRS Data

MRS data were analyzed using the LCModel software (Version 6.3-1 M) [28], which calculates the best fit as a linear combination of model spectra. The basis set comprised simulated spectra of the following metabolites: alanine, aspartate, creatine, choline (Cho), GABA, glucose, glutamine, Glu, glycerophosphorylcholine, glutathione, phosphorylcholine, phosphocreatine, scyllo-inositol, lactate, N-acetylaspartate, N-acetylaspartylglutamate, and taurine. LCModel analysis was used to quantify the concentration of neurochemicals within the chemical shift range of 0.2–4.0 ppm. Glx and Cho concentrations were automatically calculated for each voxel using LCModel. Glu is often measured as Glx (Glu plus glutamine) as the spectra overlap and are difficult to reliably distinguish [29]. All metabolite concentrations are presented as a ratio to total creatine (creatine plus phosphocreatine or tCR) as the internal reference to reduced inter-subject variability [30]. The exclusion criterion for data was as follows: Cramer–Rao Lower Bound (CRLB) value > 10%.

### 2.10. Statistics

Statistical analysis was performed using PASW software version 25 (SPSS; IBM, Armonk, NY, USA). Data normality was assessed prior to analysis using Kolmogorov-Smirnov tests, with log transformation applied when a violation was identified [31]. For clarity, data are displayed in non-transformed form [32,33].

Single-pulse MEP amplitudes and MEP amplitudes recorded under inhibitory conditions were compared by paired t-tests. Individual variability within sessions is expressed by the coefficient of variance and that between sessions by Pearson’s correlation. Inhibitory parameters (SICI, LICI, SAI, LAI), age, TMS intensity required to elicit a 1 mV MEP (TS_1 mV_), RMS, and electrical stimulation intensity were compared among BDNF genotypes by one-way ANOVA with post hoc Bonferroni tests. Glx/tCr and Cho/tCr values in each brain region were compared between the Val/Val group and Met group (consisting of both Met/Met and Met/Val carriers due to the limited number of Met/Met carriers) by independent samples t-test. The age difference between the two groups was analyzed by an independent samples t-test. In addition, Pearson’s correlation analysis was used to compare the Glx/tCr and inhibitory TMS conditions. A significant threshold of *p* < 0.05 was used.

## 3. Results

### 3.1. Participants and Measurements

No participant dropped out or presented with side effects. For 45 of the 58 participants, TS_1 mV_ (mean ± SD) was 61.5% ± 8.1%, RMT was 50.7% ± 7.5%, and electrical stimulation intensity was 10.1 ± 2.2 mA. Some MEP trials (mean ± SD per participant across two sessions) were excluded owing to the presence of movement artifacts in the EMG recordings (single-pulse, 0.63 ± 0.95 trials; SICI, 0.70 ± 1.07 trials; LICI, 0.17 ± 0.49 trials; SAI_1, 0.57 ± 0.86 trials; SAI_2, 0.63 ± 0.88 trials; LAI, 0.28 ± 0.72 trials). 

The genetic analysis classified all participants (*n* = 58) as Val/Val, Val/Met, or Met/Met genotypes (Table 1). One-way ANOVA revealed no significant associations between BDNF genotype and age, TS_1 mV_, RMT, or electrical stimulation intensity (all *p*-values > 0.08). Moreover, independent t-test revealed no significant difference in age between Met (Val/Met plus Met/Met) and Val/Val genotype groups participating in the MR experiment (*p* = 0.643). 

### 3.2. Single-Pulse MEP and Inhibitory TMS Conditions

The average single-pulse MEPs and inhibitory metrics from TMS are shown in Table 2. Paired t-tests revealed that the MEP amplitudes for each inhibitory TMS condition were reduced compared to the single-pulse MEP in session 1 (all *p*-values < 0.001), session 2 (all *p*-values < 0.001), and across both sessions (all *p*-values < 0.001). 

The coefficient of variance values was high for all inhibitory TMS conditions (SICI, 0.66; LICI, 1.11; SAI_1, 0.79; SAI_2, 0.71; LAI, 1.14). Moderate to strong positive correlations were observed for all inhibitory TMS conditions between sessions 1 and 2 (all *r*-values > 0.60, all *p*-values < 0.001).

### 3.3. Effect of BDNF Genotype on the Cortical Inhibitory Network

The effects of BDNF genotype on single-pulse MEP amplitudes and MEP ratio averages are shown in Figure 2. One-way ANOVAs revealed a significant effect of BDNF genotype on SAI_1 (*F*_(2, 42)_ = 5.358, *p* = 0.008, partial *η*^2^ = 0.203] and SAI_2 [*F*_(2, 42)_ = 6.721, *p* = 0.003, partial *η*^2^ = 0.242). Post hoc tests demonstrated that SAI_1 was significantly stronger in the Met/Met group compared to the Val/Val group (*p* = 0.010), while there is a significant trend that SAI_1 was stronger in the Val/Met group compared to the Val/Val group (*p* = 0.053). Furthermore, SAI_2 was significantly stronger in the Val/Met and Met/Met groups compared to Val/Val group (all *p*-values < 0.014). However, there was no significant difference between the Val/Met and the Met/Met groups (SAI_1 and SAI_2, all *p*-values = 1.000). One-way ANOVA revealed no significant effects of BDNF genotype on single-pulse MEP amplitude (*F*_(2, 42)_ = 1.501, *p* = 0.685, partial *η*^2^ = 0.004) or on the other inhibitory TMS conditions (all *p*-values > 0.091).

### 3.4. Effect of BDNF Genotype on Neurometabolite Concentrations

The representative voxel positions and spectra for the left M1 are presented in Figure 3. None of all participants for MRS data were removed by the exclusion criteria including FWHM and CRLB (FWHM mean ± SD: M1, 9.94 ± 1.17 Hz; S1, 9.63 ± 1.55 Hz; cerebellum, 16.52 ± 3.64 Hz; CRLB mean ± SD: Glx/tCr_M1, 5.53% ± 0.73%; Glx/tCr_S1, 5.07% ± 0.45%; Glx/tCr_cerebellum, 6.40% ± 1.10%; Cho/tCr_M1, 4.33% ± 0.55%; Cho/tCr_S1, 4.17% ± 0.38%; and Cho/tCr_cerebellum, 3.13% ± 0.51%). Among all participants, mean Glx/tCr and Cho/tCr are shown in Table 3. Glx/tCr and Cho/tCr values in Met and Val/Val groups are shown in Figure 4. The M1 Glx level was higher in the Val/Val group than the Met group (*t*_(28)_ = 2.456, *p* = 0.021, *d* = 0.899), while there were no significant differences in S1 and cerebellum (all *p*-values > 0.17). Regional Cho levels also did not differ significantly between the Val/Val and Met groups (all *p*-values > 0.32).

### 3.5. Relationship between SAI and Glx

Since BDNF genotypes only influence the SAI-related inhibitory circuit, correlation analysis was performed between SAI and Glx in each region. Positive correlation was observed between SAI_1 and the M1 Glx level (*r* = 0.485, *p* = 0.049; Figure 5). In addition, there was a likely positive correlation between SAI_2 and the M1 Glx level (*r* = 0.457, *p* = 0.065). However, there were no significant correlations between SAI_1 or SAI_2 and Glx in the others (all *p*-values > 0.38). 

## 4. Discussion

In the present study, we investigated whether BDNF genotype influences GABAergic inhibitory network activity as assessed by TMS measurements and local neurometabolite concentrations as assessed by MRS. We demonstrated three key findings: (1) SAI is stronger in Met allele carriers compared to the Val/Val group; (2) M1 Glx concentration is lower in the Met group than the Val/Val group, while S1 and cerebellar concentrations are not dependent on BDNF genotype; (3) there is a positive correlation between SAI and Glx in M1. 

Several studies have reported large inter-individual variations in SICI, with coefficient of variance values of 0.532–0.793 [5,34]. The current study found even greater coefficient of variance values (0.66 to 1.14) for all inhibitory TMS conditions. Furthermore, there were moderate to strong correlations between sessions 1 and 2 for each inhibitory TMS condition, suggesting relatively low intra-individual variability. However, a recent study reported substantial trial-to-trial variability in MEP amplitude, suggesting that a relatively large number of trials (e.g., 30) are required to reliably estimate corticospinal excitability within individuals [35]. We measured MEPs 40 times for each condition, and thus believe that our evaluation minimized the effects of trial-to-trial variability.

The present study revealed that Met carriers demonstrate stronger SAI than Val/Val carriers. The common Met polymorphism is associated with an approximately 18%–30% reduction in BDNF secretion [7], which results in both decreased cortical GABAergic activity [13] and decreased glutamatergic activity [14]. In contrast, BDNF enhances GABAergic activity in the CA1 region of rats [36]. Moreover, pharmacological studies have revealed that SAI requires GABAergic and cholinergic activity, as the muscarinic receptor antagonist scopolamine and the GABA_A_ positive modulator lorazepam both influenced SAI [37,38]. Therefore, BDNF genotype may influence SAI-related inhibitory network activity by altering GABAergic transmission.

MRS studies have found no relationship between GABA concentration and TMS measurements of GABAergic activity (i.e., SICI of ISI_1.0-3.0 ms) [15,17,18,30] and LICI [18]. In contrast, some specific TMS measurements, including the input/output curve and intracortical facilitation related to excitatory synaptic activity, are associated with MRS-assessed Glu concentration [17,30]. Moreover, a combined MRS and TMS study demonstrated that cortical silent period duration, which is thought to reflect GABAergic activity, positively correlated with M1 Glx concentration rather than GABA concentration [18]. These results imply that TMS measurements related to both excitatory and inhibitory circuits are more strongly associated with MRS-assessed Glx than MRS-assessed GABA levels. Pharmacological studies have also shown that NMDA antagonists enhance the SICI-related inhibitory network [39,40]. Furthermore, SICI is believed to reflect the sum of excitatory and inhibitory activation [41]. These findings suggest that excitatory TMS-induced synaptic activity contributes to the strength of the GABAergic inhibitory network. In addition, our results showed a reduction in M1 Glx concentration in the M1 in the Met carrier group and a unique relationship between SAI and Glx in the M1 but not S1 and cerebellum. Therefore, these results may suggest that a stronger SAI in carriers of the atypical BDNF genotype results from lower Glx concentrations in M1, and thus that lower excitatory synaptic activation may facilitate the enhancement of SAI-related GABAergic circuits.

One previous human study reported lower Glx concentrations in Met carriers but in the human hippocampus [42]. A mouse study also reported lower NMDA receptor-mediated glutamatergic activity in the hippocampus among the Met genotype group [14], consistent with our result showing lower Glx concentrations in M1 among the Met group. Our results further imply that a reduction in BDNF secretion among atypical BDNF genotype carriers contributes to reduced M1 Glx concentration. Several studies have reported M1-related behavioral differences among BDNF genotypes [9,43]. For example, BDNF genotype polymorphisms negatively influenced the effect of repetitive TMS on motor recovery in stroke patients [44]. In addition, visuomotor associative learning scores were better in the Val/Val group than the Met group [9]. Therefore, the difference in M1 Glx levels between BDNF genotypes may contribute to the observed differences in behavior. 

There was no relationship between SAI and Glx concentration in S1 or cerebellum. SAI evoked at ISI = 20 ms is generated in M1 via S1 [4,45]. Studies have previously reported a positive correlation between SAI and the N20 amplitude generated by S1 area 3b in response to peripheral electrical stimulation [45] and the presence of fiber connections between the S1 area and M1 region [46]. On the other hand, SAI induced by a slightly longer ISI (i.e., 25 ms) may be generated via cerebellum to M1 connections because transcranial direct current stimulation to the cerebellum abolished the paired associative stimulation effect for the M1 excitability change evoked by a 25-ms ISI (but not a 21.5-ms ISI) [26]. Based on these studies, we surmised that neurometabolite concentrations in S1 or the cerebellum may be important drivers of inhibitory network activity evoked by sensory input; however, inhibitory network activity was not associated with Glx concentrations in these regions. This result suggests that SAI originates from M1 via excitatory and inhibitory synaptic activities, whereas S1 and the cerebellum contribute to SAI by transmitting sensory input to M1 rather than by directly contributing to the inhibitory network.

BDNF acts as a vital trophic protein for neuronal survival [47], and also modulates cholinergic neuron activity, which is critical for cognition and may impact on mental disorders [48]. However, MRS studies have reported that BDNF genotype does not affect Cho concentration in the hippocampus [42,49]. The current results indicate that BDNF genotype also does not regulate Cho concentration in motor-related regions of the brain.

The BDNF genotype did not affect SICI, LICI, or LAI, possibly due to the involvement of distinct pathways and inhibitory networks. Similar to SAI, SICI is induced by GABAergic activity mediated by GABA_A_ receptors. However, pharmacological studies have indicated that drugs affecting distinct GABA_A_ receptor subtypes differentially influence SICI and SAI [50]. Although LICI is also induced by GABAergic activity, it depends more on the GABA_B_ receptor [51]. Recent studies have indicated that LAI is a GABA_A_-mediated process similar to SAI [27]. However, LAI was not dependent on BDNF genotype. Therefore, similar to SICI, the relevant GABA_A_ receptor subtypes may differ between LAI and SAI. Further, because LAI requires a longer ISI than SAI, LAI may be induced by different cortical regions.

We did not directly measure GABA concentrations, owing to the application of the PRESS MRS sequence, because GABA measurement generally acquired using a MEGA-PRESS sequence [52]. Therefore, we cannot confirm whether BDNF genotype affects GABA concentration. It is preferable to use a MEGA-PRESS sequence to examine the effects of BDNF genotype on GABA; however, we used a PRESS sequence with higher inter-day reliability for Glx measurement than that exhibited by JPRESS and MEGA-PRESS sequences [53] to measure Glx more appropriately. Furthermore, each group in this study, especially for the MRS study, had a small sample size, and we did not recruit the same number of participants for the experiments. Thus, a large sample size for each of the three BDNF genotypes is required for future studies.

## 5. Conclusions

Our data show that the SAI-related inhibitory network is stronger in Met allele carriers than in Val/Val carriers. On the other hand, M1 Glx concentration is a lower in Met carriers (with no difference in S1 and the cerebellum) than Val/Val carriers. These results suggest that BDNF genotype controls specific inhibitory neurotransmission and excitatory neurometabolite in the M1 only. Additionally, a lower Glx concentration in the M1 predicts stronger SAI. In other words, the Glx concentration in the M1 of Met carriers may antagonize a specific inhibitory network on the M1 through the unique relationship. These findings suggest that Met polymorphism decreases M1 plasticity, and poor motor learning, consistent with previous studies.

## Figures and Tables

**Figure 1 brainsci-11-00395-f001:**
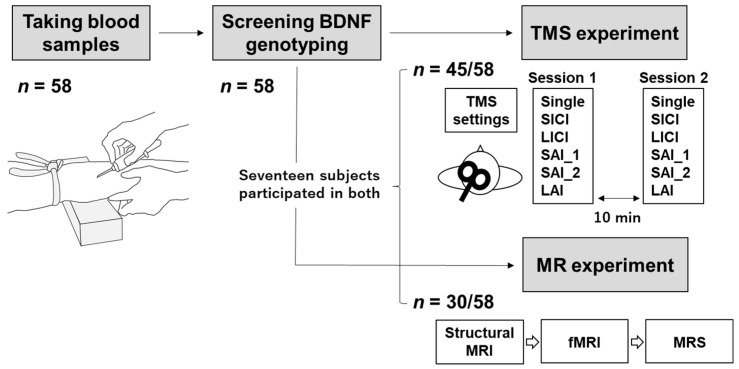
Experimental protocol. A total of 58 participants were recruited in this study. Blood samples were collected to identify the BDNF) genotype of each participant. The participants then underwent the TMS) (*n* = 45), magnetic resonance spectroscopy (MRS) experiments (*n* = 30), or both (*n* = 17). Single-pulse Motor-evoked potentials (MEPs) and inhibitory responses to TMS were measured 20 times each in two sessions, totaling 40 MEPs per condition, with 10 min intervals. The magnetic resonance (MR) data were collected in the following order: structural MRI, fMRI, and MRS. Abbreviations: BDNF, brain-derived neurotrophic factor; TMS, transcranial magnetic stimulation; Single, single-pulse; SICI, short-interval intracortical inhibition; LICI, long-interval intracortical inhibition; SAI_1, short-latency afferent inhibition with an ISI of 20 ms; SAI_2, short-latency afferent inhibition with an ISI of 25 ms; LAI, long-latency afferent inhibition; MR, magnetic resonance; MRI, magnetic resonance imaging; fMRI, functional magnetic resonance imaging; MRS, magnetic resonance spectroscopy.

**Figure 2 brainsci-11-00395-f002:**
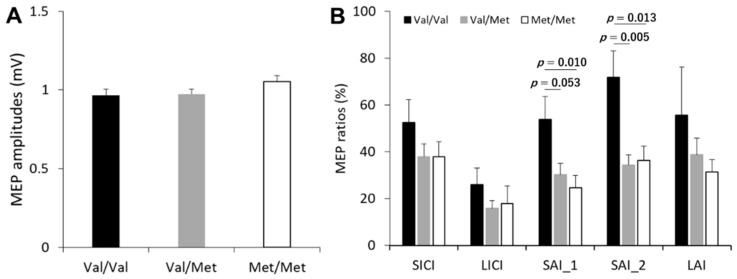
Effect of BDNF genotype on cortical inhibitory networks as assessed by TMS. (**A**) Average single-pulse MEP amplitude in each group of BDNF genotype. (**B**) Average MEP ratio for each inhibitory TMS condition by BDNF genotype. Abbreviations: MEP, motor-evoked potential; SICI, short-interval intracortical inhibition; LICI, long-interval intracortical inhibition; SAI_1, short-latency afferent inhibition with an ISI of 20 ms; SAI_2, short-latency afferent inhibition with an ISI of 25 ms; LAI, long-latency afferent inhibition.

**Figure 3 brainsci-11-00395-f003:**
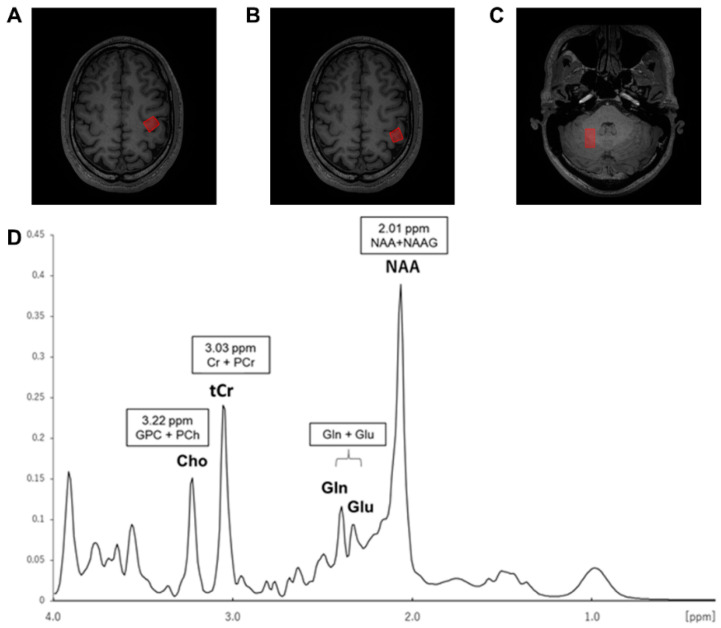
Overview of the voxel positions and representative spectrum obtained from the left M1. (**A**) Left M1. (**B**) left S1. (**C**) right cerebellum. (**D**) A representative spectrum acquired from the left M1 with the point resolved spectroscopy sequence (PRESS) at 3T MRS. Abbreviations: ppm, parts per million; GPC + PCh, glycerophosphocholine plus phosphocholine; Cho, choline; Cr plus PCr, creatine plus phosphocreatine; tCr, total creatine; Gln, glutamine; Glu, glutamate; NAA, N-acetyl aspartate; NAAG, N-acetylaspartylglutamate.

**Figure 4 brainsci-11-00395-f004:**
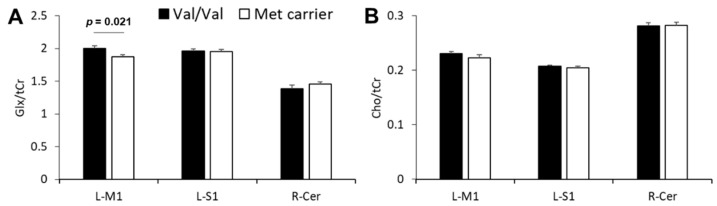
Effect of BDNF genotype on Glx and Cho levels. (**A**) Average Glx/tCr levels for the left M1 (L-M1), left S1 (L-S1), and right cerebellum (R-Cer) regions by BDNF genotypes, the Val/Val group or Met group. (**B**) Average Cho/tCr levels for each region by BDNF genotypes. The Met group included both the Val/Met (*n* = 11) and Met/Met (*n* = 5) carriers. Abbreviations: Glx, glutamate plus glutamine; tCr, total creatinine; L-M1, left primary motor cortex; L-S1, left primary somatosensory cortex; R-Cer, right cerebellum; Cho, choline.

**Figure 5 brainsci-11-00395-f005:**
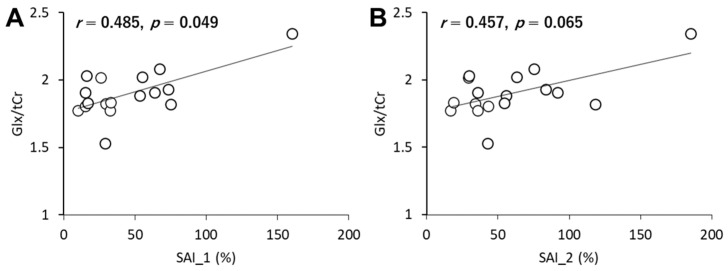
Association between SAI and Glx in the M1. (**A**) SAI_1 and Glx. (**B**) SAI_2 and Glx.

**Table 1 brainsci-11-00395-t001:** Participant information and TMS settings by BDNF genotype in TMS and MR experiments.

	Val/Val	Val/Met	Met/Met
**TMS (*n* = 45)**			
Participants	14	19	12
Age	21.9 ± 1.7	21.8 ± 1.3	22.8 ± 3.0
TS_1 mV_	61.7 ± 8.3	63.1 ± 7.3	58.7 ± 9.0
RMT	48.9 ± 9.1	52.8 ± 6.3	49.3 ± 7.1
Electrial stimulation	10.3 ± 2.2	10.7 ± 2.1	9.0 ± 2.0
**MR (*n* = 30)**			
Participants	14	11	5
Age	22.1 ± 2.1	22.2 ± 1.7	21.6 ± 0.5

Values are expressed as mean ± SD. Abbreviations: TMS, transcranial magnetic stimulation; TS_1 mV_, transcranial magnetic stimulation intensity required to elicit a 1 mV MEP; RMT, resting motor threshold; MR, magnetic resonance.

**Table 2 brainsci-11-00395-t002:** Mean MEP amplitudes for single-pulse MEP and inhibitory TMS conditions.

	Both Sessions	Session 1	Session 2
Single-pulse	0.99 ± 0.02	1.01 ± 0.02	0.98 ± 0.02
SICI	0.43 ± 0.04	0.44 ± 0.05	0.41 ± 0.04
LICI	0.19 ± 0.03	0.20 ± 0.04	0.19 ± 0.03
SAI_1	0.36 ± 0.05	0.36 ± 0.04	0.36 ± 0.06
SAI_2	0.47 ± 0.05	0.48 ± 0.06	0.45 ± 0.06
LAI	0.43 ± 0.08	0.43 ± 0.07	0.43 ± 0.09

Values are expressed as mean ± SEM. Abbreviations: SICI, short-interval intracortical inhibition; LICI, long-interval intracortical inhibition; SAI_1, short-latency afferent inhibition with an ISI of 20 ms; SAI_2, short-latency afferent inhibition with an ISI of 25 ms; LAI, long-latency afferent inhibition.

**Table 3 brainsci-11-00395-t003:** Mean neurometabolite concentrations in M1, S1, and the cerebellum.

	Left M1	Left S1	Right Cerebellum
Glx/tCr	1.93 ± 0.03	1.96 ± 0.02	1.42 ± 0.03
Cho/tCr	0.23 ± 0.00	0.21 ± 0.00	0.28 ± 0.00

Values are expressed as mean ± SEM. Abbreviations: Glx, glutamate plus glutamine; tCr, total creatinine; Cho, choline; M1, primary motor cortex; S1, primary somatosensory cortex.

## Data Availability

Data in this study will be made available upon request to the corresponding author.

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
