# Peer review of "Influence of Brain-Derived Neurotrophic Factor Genotype on Short-Latency Afferent Inhibition and Motor Cortex Metabolites"

_brainsci, 2021, doi:10.3390/brainsci11030395_

Round 1
Reviewer 1 Report
The reviewed paper presents the relationship between BDNF genotype and GABAergic inhibitor network activity as assessed by TMS measurements and local neurometabolite concentrations as assessed by MRS. The manuscript is written correctly and clearly. The study is interesting, although similar reports on a similar group size are already published.
Sugestions:
- title should be corrected. The study did not rather investigate the influence of BDNF on short-latency afferent inhibition and motor cortex metabolites, but rather the relationship.
- introduction of BDNF is laconic. The importance of BDNF should be described in more detail, especially in neurological, psychiatric and immune-related diseases (10.1111/ene.12976; 10.1038/aps.2010.184; 10.1111/nmo.13978; 10.1016/j.pnpbp.2017.11.006)
- the introduction should be written clearer; especially the importance of the topic
- the study group is quite small for a genetic test; this was not highlighted in the discussion as limitations
- if all the data are normally distributed? For example, in a group of 12 participants this is quite unlikely.
- why not all patients underwent MRI and TMS?
- figure 1 suggests that one group underwent TMS due to BDNF genotype and another group underwent the MR, which is probably not true. The division of groups is very unclear.
- abbreviations for tables and figures are not explained
- single abbreviations in the text are unexplained
- the conclusion should not repeat the purpose of the study. The conclusions should be transparent and clear.
Author Response
Comments and Suggestions for Authors
The reviewed paper presents the relationship between BDNF genotype and GABAergic inhibitor network activity as assessed by TMS measurements and local neurometabolite concentrations as assessed by MRS. The manuscript is written correctly and clearly. The study is interesting, although similar reports on a similar group size are already published.
Suggestions:
- title should be corrected. The study did not rather investigate the influence of BDNF on short-latency afferent inhibition and motor cortex metabolites, but rather the relationship.
Comment 1; We changed the title to be clearer (page. 1).
- introduction of BDNF is laconic. The importance of BDNF should be described in more detail, especially in neurological, psychiatric and immune-related diseases (10.1111/ene.12976; 10.1038/aps.2010.184; 10.1111/nmo.13978; 10.1016/j.pnpbp.2017.11.006)
Comment 2; Thank you for providing us with these research articles regarding BDNF. We added some sentences and these research articles to explain it in detail (page. 2, lines 60-62).
- the introduction should be written clearer; especially the importance of the topic
Comment 3; We revised some sentences in the introduction to be clearer the main topic (pages. 2-3, lines 42-114).
- the study group is quite small for a genetic test; this was not highlighted in the discussion as limitations
Comment 4; We added it in the limitation section (page. 12, lines 483-486).
- if all the data are normally distributed? For example, in a group of 12 participants this is quite unlikely.
Comment 5; In MRS experiment, some non-distributions were observed only Cho concentrations. Therefore, we decided to use log transformation to close to normal distribution [1], and all data were analyzed by parametric test with log transformation (page. 6, lines 251-254; pages. 6-7, lines 282-372).
- why not all patients underwent MRI and TMS?
Comment 6; We think that recruiting same participants between TMS and MRS experiments are the best way. However, we could not recruit all participants by several reasons, such as graduation, long-term clinical training, needed experimenter who is qualified for a national license, and so on. We described it as a limitation (page. 12, lines 483-486).
- figure 1 suggests that one group underwent TMS due to BDNF genotype and another group underwent the MR, which is probably not true. The division of groups is very unclear.
Comment 7; Both experiments included some participants who were genotyped from 58 participants (Fig. 1; page. 3, lines 119-124).
- abbreviations for tables and figures are not explained
- single abbreviations in the text are unexplained
Comment 8; We added abbreviations each table and figure.
- the conclusion should not repeat the purpose of the study. The conclusions should be transparent and clear.
Comment 9; We deleted the purpose of the study and revise it to be transparent and clear (Page. 12-13, lines 489-502).
Reviewer 2 Report
Sasaki and colleagues investigate the influence of the met allele of the BDNF gene on a variety of inhibitory TMS measures that are reflective of GABAergic activity within M1. Additionally, they used MRS to measure potential allele-related differences in the dynamics of glutamate/glutamine (Glx) and choline concentrations in M1, S1, and the cerebellum. They found that individuals with a met allele (met/met or val/met) had stronger SAI than those without (val/val). The met group also had less Glx in M1. The study is interesting and provides a useful addition to the current literature. I have a few comments below.
Introduction
The introduction provides clear and useful background information. However, in the final section, I would like to see the research question more explicitly laid out before getting into the hypotheses. The overarching research question seems to be to understand the impact of BDNF genotype on inhibition within the motor cortex broadly, with TMS and MRS providing the means through which the question can be investigated. It would be useful to understand whether these two are expected to provide unique information or if a relationship between the two is expected or will be tested. Potential explanations for these things are found in the background information, but not in the research question itself.
Methods
Section 2.10: For the paired t-tests between the conditioned and unconditioned responses used to determine whether inhibition was occurring, were the allele groups pooled together? Is it possible that there would not be inhibition in one group and therefore they should be assessed with an ANOVA that includes group as a factor?
Just to clarify, for the TMS measures, age, RMS, and electrical stimulation intensity, an ANOVA was conducted because there were three groups (met/met, val/met, and val/val) whereas t-tests were used for the MRS measures because the met groups were collapsed together?
If this is true, do the results hold if the TMS measures are looked at with the met groups combined? If there is physiological rationale for combining the met groups, as was done in the MRS, I think this should be looked at. Otherwise, if there is not physiological rationale, then it doesn’t seem like a useful approach in the MRS data.
Discussion
The authors talk about other studies that looked at correlations between MRS and TMS measures. Why was this not done in the current work? This might be relevant, especially given that the authors make conclusions such as “These results may suggest that a strong SAI in carriers of the atypical BDNF genotype results from lower Glx concentrations in M1…” and talk about there being no relationship between SAI and Glx in S1 or cerebellum. While it’s true that Glx in these locations didn’t change, the actual relationships between TMS and MRS measures do not seem to have been tested.
Author Response
Comments and Suggestions for Authors
Sasaki and colleagues investigate the influence of the met allele of the BDNF gene on a variety of inhibitory TMS measures that are reflective of GABAergic activity within M1. Additionally, they used MRS to measure potential allele-related differences in the dynamics of glutamate/glutamine (Glx) and choline concentrations in M1, S1, and the cerebellum. They found that individuals with a met allele (met/met or val/met) had stronger SAI than those without (val/val). The met group also had less Glx in M1. The study is interesting and provides a useful addition to the current literature. I have a few comments below.
Introduction
The introduction provides clear and useful background information. However, in the final section, I would like to see the research question more explicitly laid out before getting into the hypotheses. The overarching research question seems to be to understand the impact of BDNF genotype on inhibition within the motor cortex broadly, with TMS and MRS providing the means through which the question can be investigated. It would be useful to understand whether these two are expected to provide unique information or if a relationship between the two is expected or will be tested. Potential explanations for these things are found in the background information, but not in the research question itself.
Comment 1; We added some sentences in the last paragraph (page. 3, lines 100-104).
Methods
Section 2.10: For the paired t-tests between the conditioned and unconditioned responses used to determine whether inhibition was occurring, were the allele groups pooled together? Is it possible that there would not be inhibition in one group and therefore they should be assessed with an ANOVA that includes group as a factor?
Just to clarify, for the TMS measures, age, RMS, and electrical stimulation intensity, an ANOVA was conducted because there were three groups (met/met, val/met, and val/val) whereas t-tests were used for the MRS measures because the met groups were collapsed together?
If this is true, do the results hold if the TMS measures are looked at with the met groups combined? If there is physiological rationale for combining the met groups, as was done in the MRS, I think this should be looked at. Otherwise, if there is not physiological rationale, then it doesn’t seem like a useful approach in the MRS data.
Comment 2; We pooled all groups when analyzed by t-test. Does it mean that we should use ANOVA that includes BDNF genotype group as a factor each inhibitory TMS condition? If so, we used ANOVA to confirm BDNF genotype-related changes in each paired pulse condition. In addition, we used t-statistics to know whether inhibition was occurred in all participants without BDNF genotype group because we needed to confirm that all inhibitory paradigms suppressed MEP as well as previous studies.
We used t-test to compare between Val/Val and Met carriers (i.e., Val/Met and Met/Met) in MRS experiment (page. 6, lines 263-264). On the other hand, ANOVA were conducted for three groups (i,e,. Val/Val, Val/Met, and Met/Met) in TMS experiment.
The results did not change when combined Val/Met and Met/Met as Met carriers each TMS condition using an independent t-test (single-pulse, p = 0.415; SICI, p = 0.105; LICI, p = 0.184; SAI_1, p = 0.004; SAI_2, p < 0.001; LAI, p = 0.203). The population of Met/Met is much less than the others, and it was investigated all over the world (Met/Met, 2.0%-23.4%) [2]. Therefore, Met carriers combined Val/Met and Met/Met are often used [3-6]. However, it is a limitation in our study, we described it in the limitation section (page. 12, lines 483-486).
Discussion
The authors talk about other studies that looked at correlations between MRS and TMS measures. Why was this not done in the current work? This might be relevant, especially given that the authors make conclusions such as “These results may suggest that a strong SAI in carriers of the atypical BDNF genotype results from lower Glx concentrations in M1…” and talk about there being no relationship between SAI and Glx in S1 or cerebellum. While it’s true that Glx in these locations didn’t change, the actual relationships between TMS and MRS measures do not seem to have been tested.
Comment 3; We analyzed the correlations between SAI and Glx in each cortical region. Therefore, we added some sentence regarding the correlation in abstract (page. 1, lines 31-33), method (page. 6, lines 264-266), results (page. 10, lines 366-372; figure. 5) and discussion (page. 11, lines 417-419).
References;
- Feng, C., Wang, H., Lu, N., Tu, X.M. Log transformation: application and interpretation in biomedical research. Stat Med. 2013, 32, 230-9.
- Shen, T., You, Y., Joseph, C., Mirzaei, M., Klistorner, A., Graham, S.L., Gupta, V. BDNF Polymorphism: A Review of Its Diagnostic and Clinical Relevance in Neurodegenerative Disorders. Aging Dis. 2018, 9, 523-36.
- Witte, A.V., Kurten, J., Jansen, S., Schirmacher, A., Brand, E., Sommer, J., Floel, A. Interaction of BDNF and COMT polymorphisms on paired-associative stimulation-induced cortical plasticity. J Neurosci. 2012, 32, 4553-61.
- Cheeran, B., Talelli, P., Mori, F., Koch, G., Suppa, A., Edwards, M., Houlden, H., Bhatia, K., Greenwood, R., Rothwell, J.C. A common polymorphism in the brain-derived neurotrophic factor gene (BDNF) modulates human cortical plasticity and the response to rTMS. J Physiol. 2008, 586, 5717-25.
- Taschereau-Dumouchel, V., Hetu, S., Michon, P.E., Vachon-Presseau, E., Massicotte, E., De Beaumont, L., Fecteau, S., Poirier, J., Mercier, C., Chagnon, Y.C., Jackson, P.L. BDNF Val(66)Met polymorphism influences visuomotor associative learning and the sensitivity to action observation. Sci Rep. 2016, 6, 34907.
- Marsili, L., Suppa, A., Di Stasio, F., Belvisi, D., Upadhyay, N., Berardelli, I., Pasquini, M., Petrucci, S., Ginevrino, M., Fabbrini, G., Cardona, F., Defazio, G., Berardelli, A. BDNF and LTP-/LTD-like plasticity of the primary motor cortex in Gilles de la Tourette syndrome. Exp Brain Res. 2017, 235, 841-50.